# Can Polylactic Acid (PLA) Act as an Important Vector for Triclosan?

**Zihan Lang [1,\*] and Lidong Xue [2,\*]**

1   College of Resources and Environment, Shandong Agricultural University, Tai'an 271018, China
2   China National Environmental Monitoring Centre, Beijing 100012, China
\*   Correspondence: langzihan2011@126.com (Z.L.); xueld@cnemc.cn (L.X.)

**Abstract:** Triclosan (TCS) has been widely employed as active ingredient in household products and has received attention for its hepatotoxicity, endocrine disruption and disturbance on immune function. Polylactic acid (PLA) has been highlighted as an alternative biodegradable microplastic, and the knowledge about the adsorption affinity towards TCS is limited. In this study, the ability to act as carrier of TCS by PLA and non-biodegradable microplastics, including polystyrene (PS), polyvinyl chloride (PVC) and polyethylene (PE) with different particle sizes were explored. The influence factors (e.g., dosage, pH and salinity), adsorption kinetics and isotherms were also investigated. Batch experiment results indicated that the TCS adsorption onto PLA and non-biodegradable microplastics exhibited a pronounced pH-dependent pattern (pH of 4, 7 and 10), and the adsorption capacity decreased gradually as pH increased. Furthermore, the adsorption capacity of TCS on PS, PVC and PE decreased as salinity increased from 0 to 3.5%, while no significant inhibition for the sorption capacity of PLA was observed. The adsorption kinetic data of TCS was best fitted with the pseudo-second order model. The Freundlich model with $R^2$ (0.999) was suitable to describe the adsorption isotherms of TCS on PLA, while the isotherms data of TCS on PS, PVC and PE was fitted by linear and Freundlich model. The higher adsorption capacity of PLA (38.6 mg g$^{-1}$) compares to those of PS, PVC and PE (31.3, 11.4 and 9.64 mg g$^{-1}$, respectively), illustrated by the fact that the physicochemical properties of microplastics have a noticeable impact on adsorption process, and the biodegradable PLA is a stronger vector than the non-biodegradable microplastics.

**Keywords:** triclosan; polylactic acid; vector; pH-dependent

## 1. Introduction

Plastics are organic polymers that produced from monomers utilized addition polymerization or condensation polymerization [1,2]. The rapid development of the plastics industry has brought convenience to people's lives, as well as a series of environmental problems caused by waste plastics pollution. Microplastics are generally defined as plastic with particles less than 5 mm in diameter [3]. Previous studies revealed that the amount of microplastics entering soil, ocean is over $7 \times 10^5$ t a$^{-1}$ and $9.3 \times 10^4$ t a$^{-1}$, respectively [1,3]. Microplastics can remain in the environment for a long time due to their relatively stable chemical properties and extremely slow degradation process. Microplastics, as emerging pollutants, can act as vectors for contaminants, which have drawn more attention. The higher accumulations of heavy metals and organic compounds by microplastics have been reported [3,4]. A previous study demonstrated that the adsorption ability of organic pollutants onto microplastics was affected by its properties and solution chemistry [4]. The sorption behavior and process can be influenced by the properties of the sorbent and sorbate [5–7]. For instance, the functional groups, degree of crystalline and polarity of microplastics play a crucial part in environmental pollutants adsorption [4,6].

In general, microplastics fall into two categories, non-biodegradable and biodegradable, depending on their completely degraded ability by microorganisms. Biodegradable microplastics are increasingly used to reduce the environmental impact of large amounts of residual non-biodegradable microplastics in the environment. As an alternative to

conventional microplastic, polylactic acid (PLA), first discovered in 1932, is one of the biodegradable microplastics that is widely used (e.g., biomedical applications) to replace the non-biodegradable materials [8]. With the increasing demand for biodegradable PLA, the production capacity of PLA is predicted to reach around 800,000 tons in 2020 [9]. As large quantities of biodegradable microplastics are introduced into the environment, the coexistence and interactions between biodegradable microplastics and contaminants are becoming a hot topic of concern. Zuo et al. found that the adsorption capacity of phenanthrene on biodegradable PBAT was higher than on polyethylene (PE) and polystyrene (PS) [10]. However, the knowledge of adsorption affinity of PLA is limited.

Triclosan (TCS), a kind of broad-spectrum antiseptic used in pharmaceutical and personal care products, has been widely employed as active ingredients in household products (e.g., toothpastes, mouthwash, soap, cosmetics, detergents, medical disinfection and health care products) [11]. Owing to the massive application of TCS and its incomplete elimination, TCS ultimately was imported into environmental matrix and was detected in drinking water, groundwater, seawater, and even in the tissues and organs of living organisms [11,12]. The effects of TCS on hepatotoxicity, endocrine and immune function have been reported [13,14]. There has been growing concern about the potential negative effects of TCS in the environment on human health. The adsorption approach developed posing the potential to eliminate TCS residues and minimize its threatening risks. The high accumulation of environmental pollutants on non-biodegradable microplastics was discussed in previous studies [15,16]. Recently, adsorption process of triclosan on non-biodegradable and biodegradable microplastics was reported [17,18]. However, the study by Webb et al. [17] mainly focused on the adsorption of triclosan by non-biodegradable PE. The influence of particle size and dosage were not investigated in PLA adsorption TCS by Shi et al. [18], and the interaction of biodegradable PLA with TCS remains unclear. It is of paramount significance to conduct a series of studies evaluating the adsorption vector for TCS by PLA.

In this paper, a widely used biodegradable PLA and non-biodegradable microplastics (PE, PVC and PS) were chosen as the adsorbent to adsorb TCS. The objective of this study was: (1) to analyze and compare the PLA ability to act as a vector for TCS; (2) to demonstrate the influence factors of particle size, dosage, pH and salinity affecting the adsorption capacity; and (3) to investigate the adsorption kinetics and isotherm, revealing the adsorption behavior.

## 2. Materials and Methods

### 2.1. Materials

PLA and non-biodegradable microplastics, named as PE, PVC and PS, were purchased as powders from Shanghai Aladdin Bio-Chem Technology Co., Ltd. (Shanghai, China). The molecular structure of PLA, PE, PVC and PS are presented in Table S1. The TCS (purity of 97%) was supplied by the Shanghai Aladdin Bio-Chem Technology Co., Ltd. (Shanghai, China). TCS stock solutions were prepared with ultra-pure water by adding 10% ethanol, stored at 4 °C and used within a week of preparation. The HPLC grade methanol and acetonitrile were used for mobile phases obtained from Sigma-Aldrich (Shanghai, China). All other chemicals were of analytical grade and commercially available.

### 2.2. Batch Adsorption Experiments

A series of adsorption experiments were conducted at 25 °C. For adsorption isotherm experiments, 30 mL of TCS concentrations (from 2 to 20 mg L$^{-1}$) were set in 50 mL Erlenmeyer flasks with 15 mg of microplastics (the solid/solution ratio of 15 mg 30 mL$^{-1}$). At the same condition as isotherm, the solution with an initial TCS concentration of 8 mg L$^{-1}$ was used for the adsorption kinetic and the remaining TCS concentration was detected at different time intervals within 2160 min. All the vials were shaken at 190 rpm to reach apparent equilibrium under the darkness condition. After equilibration, the supernatant

was filtered through a 0.45 μm filtration membrane, and the concentrations of TCS were measured by HPLC.

The effects of the addition amounts of microplastics were investigated by adding different concentrations (from 0 to 2.0 g/L) of PLA, PE, PVC and PS to solutions with the same initial TCS concentration as the adsorption kinetic. To figure out the effects of chemistry conditions in solution, single-factor experiments were conducted with 8 mg $L^{-1}$ of TCS and the solid/solution ratios were the same as for the kinetic and isotherm. The pH effect experiments were conducted at the solution pH of 4, 7 and 10, adjusted by 0.1 M HCl or NaOH, respectively. The influences of the salinity effect were examined in the content of NaCl from 0 to 3.5% under the pH of 4. The experiments were performed in triplicate with similar results.

### 2.3. TCS Determination

The TCS concentration was detected by an HPLC (Shimadzu, LC-10A, Kyoto, Japan) coupled with a diode array detector (wavelength of 199 nm) and a C18 reversed phase column (5 μm × 150 mm × 4.6 mm). The mobile phase was acetonitrile: deionized water (75:25, *v*/*v*) at the flow rate of 1 mL/min. The injection volume of the sample was 10 μL and the column temperature was set at 30 °C.

### 2.4. Data Analysis

Based on the measured concentration of TCS, the removal efficiency η (%) and the adsorption capacity $Q_t$ (mg $g^{-1}$) of microplastics were calculated according to the formulas below:

$$\eta = \frac{C_0 - C_e}{C_0} * 100 \tag{1}$$

$$Q_t = \frac{(C_0 - C_e) * V}{m} \tag{2}$$

where $C_0$ (mg $L^{-1}$) represents the initial concentrations of TCS; $C_e$ (mg $L^{-1}$) represents the equilibrium concentrations of TCS; m (g) is the quality of microplastics; and the solution volume is represented by V (L).

The commonly used adsorption kinetic models described in previous studies [19,20] were as follows:

Pseudo-first-order model:

$$Q_t = Q_e * \left(1 - e^{\frac{1}{K_1 * t}}\right) \tag{3}$$

Pseudo-second-order model:

$$Q_t = \frac{K_2 * Q_e^2 * t}{(1 + K_2 * Q_e * t)} \tag{4}$$

where $K_1$ (min $^{-1}$) and $K_2$ [g (mg min) $^{-1}$] are the equilibrium rate constants of pseudo-first-order and pseudo-second-order models, respectively; t (min) is the contacting time; $Q_e$ (mg $g^{-1}$) is the sorption capacity at equilibrium.

Adsorption isotherms were described as follows:

Freundlich model:

$$Q_e = K_F * (C_e)^n \tag{5}$$

Langmuir model:

$$Q_e = \frac{Q_{max} * K_L * C_e}{(1 + K_L * C_e)} \tag{6}$$

The linear model:

$$Q_e = K_d * C_e \tag{7}$$

where $K_F$ ($(mg\ kg^{-1})\ (mg\ L^{-1})^{-n}$) and n are the adsorption coefficient and exponent of Freundlich isotherm model; $Q_{max}$ ($mg\ g^{-1}$) stands for the maximum adsorption capacity; $K_L$ ($L\ mg^{-1}$) is the adsorption coefficient of Langmuir isotherm model; and $K_d$ ($L\ g^{-1}$) is the distribution coefficient of the Linear isotherm model.

### 3. Results and Discussion

*3.1. The Influence of Particle Size of Microplastics*

The particle size of microplastics has significant influence on the adsorption affinity [15,16,20]. After grinding and sieving, the particle size of microplastics was divided from 75 μm to 150 μm (75–150 μm account for 70% among them). The particle sizes distribution of PLA, PE, PVC and PS are shown in Figure S1. The effect of microplastic particle size on the TCS adsorption process was investigated at the initial concentration of TCS of 8 mg $L^{-1}$. The results in this experiment showed that the adsorption capacity of PLA with different particle size was higher than that of PE, PVC and PS, which revealed that PLA was a stronger carrier for TCS. With the increase of particle size, the Qt value nearly remained the same. The influence of particle sizes on adsorption capacity was not obvious (Figure 1). The adsorption properties of microplastics increased with the decrease in particle size have been found in previous studies [14,21]. The inconsistent results might be attributed to the different particle sizes used in this experiment.

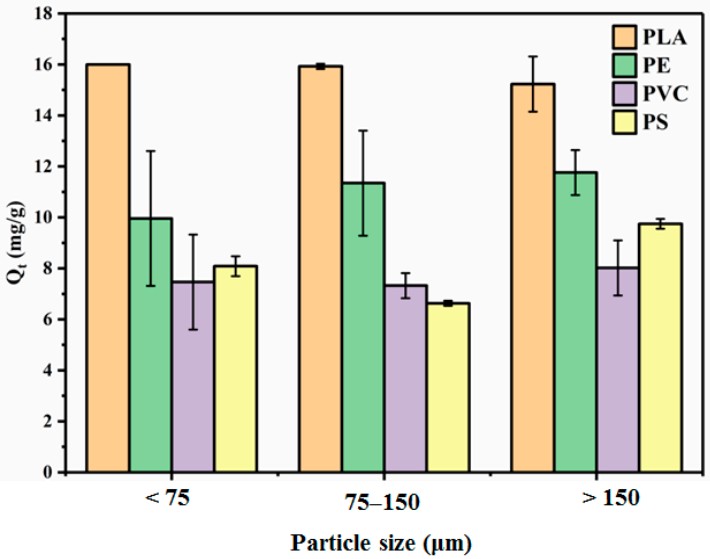

**Figure 1.** Effect of particle size on the sorption of TCS on PLA, PE, PVC and PS.

*3.2. The Influence of Microplastics Dosage*

The additive amount of microplastics is also a critical influence factor for TCS adsorption processes. Subsequently, the influence of microplastics dosage on the adsorption of TCS was studied. As shown in Figure 2, TCS was completely removed from the solution in which PLA was added, and the removal efficiency (η) remained at 100% with the increased dosage of PLA from 0.5 g/L to 2 g/L. For PE, PVC and PS, the η of TCS increased with the increase of microplastics dosage, respectively. Among them, the η of PE is above 75%. The removal efficiencies of the PVC and PS are similar, ranging from 30% to 60%. When the microplastics dosage is low, TCS can completely occupy the adsorption site on the surface of microplastics. The increase of adding dosage can contribute to a substantial increase of the removal efficiency by providing more available adsorption sites [15,22–24].

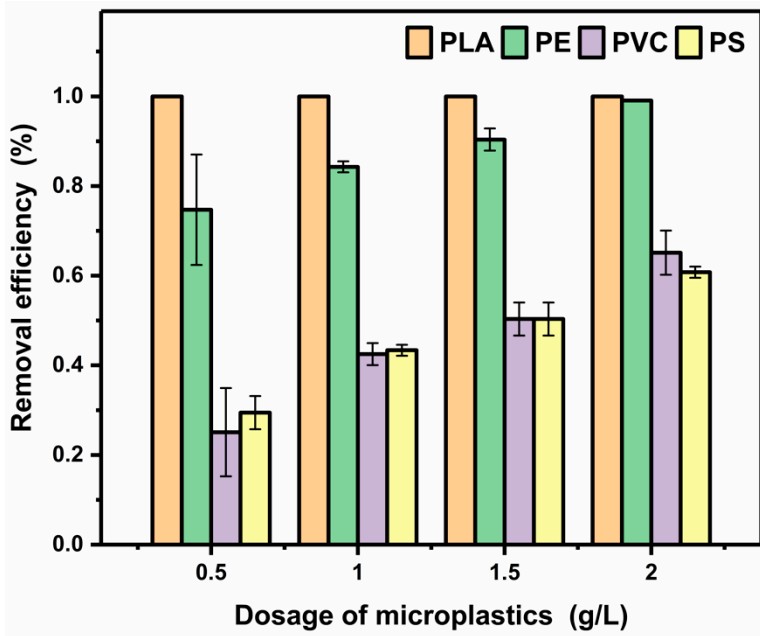

**Figure 2.** The influence of microplastics dosages on removal efficiency of TCS.

### 3.3. The Influence of pH

The effect of pH on the adsorption of TCS by PLA, PE, PVC and PS was shown in Figure 3, and the adsorption capacity of the TCS decreased as pH increased. Under different pH conditions, changes in both the physio-chemical properties of TCS and the surface charge of microplastics lead to the change of adsorption performance [18,25]. On the one hand, as an ionizable organic chemical, the dissociation of TCS molecules can be affected by the pH value. The distribution of dissociated and non-dissociated species of TCS with pH values is represented in Figure S2. The dissociation constant ($pK_a$) of TCS is 8.14, when pH < pKa, TCS is mainly distributed in solution in non-dissociated form in solution; whereas when pH > $pK_a$, the dissociated species (anions) is dominant. On the other hand, regarding microplastics, the surface functional groups are deprotonated when the solution pH is higher than the $pH_{pzc}$ of the four microplastics. Therefore, the microplastics surface is negatively charged at solution pH > $pH_{pzc}$ [23,25,26]. The point of zero charge (PZC) was determined by the titration method proposed by Tripathy et al. (measurement method see Supplementary Materials) [27]. The $pH_{pzc}$ of PLA, PE, PVC and PS is 7.3, 7.0, 7.4 and 7.2, respectively. The surface of the microplastics and TCS molecules became more negatively charged with the increase of solution pH. The primary interaction between TCS and the surface of microplastics is electrostatic repulsion, which hinders the adsorption of TCS with negative charges. The results indirectly indicated that the electrostatic interaction plays a vital part in the potential of microplastics as a vector for TCS.

### 3.4. The Influence of Salinity

Salinity is an important factor that potentially influences the contaminants adsorption on microplastics. The adsorption capacity of TCS by non-biodegradable PE, PVC and PS was decreased as NaCl concentrations increased. However, the variation in PLA adsorption at different NaCl concentrations was not noticeable (Figure 4). To a certain extent, the rise of ionic strength will lead to the inhibition of the adsorption capacity of certain contaminants on microplastics [28]. When ionic strength increases, the absorbed $Na^+$ on the surface of microplastics will compete with adsorbents for adsorption sites. In addition, exchangeable inorganic cation ($Na^+$) can easily replace the hydrogen ion, thus inhibiting the synthesis of hydrogen bond. In the study, the adsorption capacity of microplastics was not obviously decreased as salinity increased in the range of 0.1% to 3.5%. The reasons might be attribute to: (1) TCS solubility in the aqueous solution decreased with

the increase of salinity, the quantity of the dissolved organic decreased and the quantity of solid phase organic increased; (2) the formed suspension-colloid system at adsorption equilibrium was destroyed and flocculation occurred as the salinity increased [16,28]. The salinity showed no obvious influence on the potential of PLA as a carrier for TCS, indicating that the competitive adsorption between $Na^+$ and TCS on the surface of PLA was complicated, and the potential interaction still requires further study.

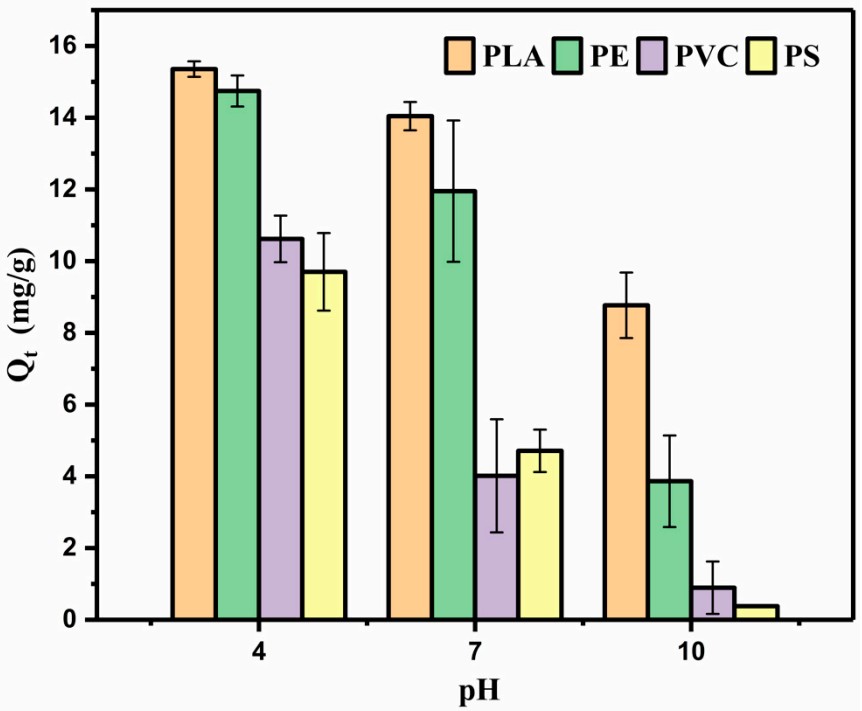

**Figure 3.** Effect of pH on TCS adsorption performance onto PLA, PE, PVC and PS.

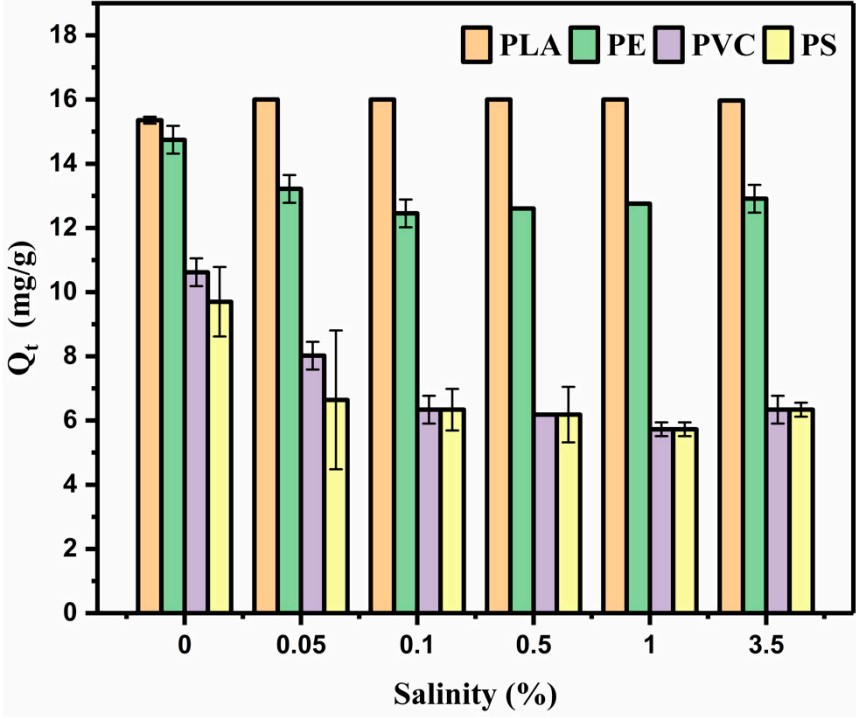

**Figure 4.** Effect of salinity on TCS adsorption onto PLA, PE, PVC and PS.

### 3.5. Adsorption Kinetics

The adsorption kinetics of TCS on PLA, PE, PVC and PS are presented in Figure 5. From the viewpoint of the overall adsorption procedure, the adsorption TCS was rapid in the initial stage with large accessible active sites on microplastics, and then gradually reached adsorption equilibrium with binding sites occupied and the equilibrium time of 24 h. In order to pursue the rate-limiting mechanisms of TCS adsorption on PLA, PE, PVC and PS, the kinetics for adsorption were described by the pseudo-first order and pseudo-second order model, and the dynamic parameters were listed in Table 1. The calculated $Q_e$ values follow an order of PLA > PE > PVC > PS, which is in accordance with the experimental results. Therefore, the results manifested that the adsorption of TCS was significantly influenced by the characteristics of biodegradable and non-biodegradable microplastics. The coefficient ($R^2$) of the pseudo-second order model for PLA, PE, PVC and PS was larger than that of the pseudo-first order model, respectively. Moreover, there was a close agreement between the experimentally measured $Q_e$ values and the calculated $Q_e$ values obtained from the pseudo-second order model. Based on the above analysis, the pseudo-second order model was more suitable for fitting the TCS kinetic process. A similar result has been found in Li et al.'s [21] investigation of the diclofenac adsorption by PS. The adsorption kinetics of TCS onto PLA, PE, PVC and PS indicated that chemical force was the main controlling element, and chemisorption was the dominating rate-limiting step.

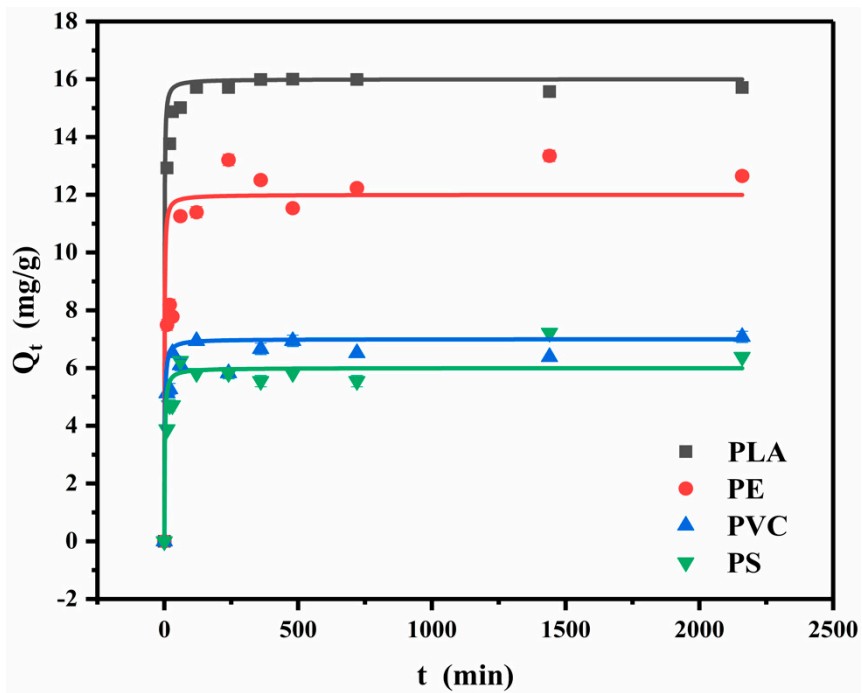

**Figure 5.** The pseudo-second order model of TCS on PLA, PE, PVC and PS.

**Table 1.** Adsorption kinetics of TCS onto PLA, PE, PVC and PS.

| Kinetics Model | Parameters | PLA | PE | PVC | PS |
|---|---|---|---|---|---|
| pseudo-first order | $K_1$ (h$^{-1}$) | 0.159 | 0.055 | 0.130 | 0.082 |
| | $Q_e$ (mg g$^{-1}$) | 15.6 | 12.2 | 6.52 | 5.99 |
| | $R^2$ | 0.987 | 0.914 | 0.948 | 0.917 |
| pseudo-second order | $K_2$ (g mg$^{-1}$ h$^{-1}$) | 0.026 | 0.008 | 0.046 | 0.026 |
| | $Q_e$ (mg g$^{-1}$) | 15.9 | 12.7 | 6.67 | 6.18 |
| | $R^2$ | 0.998 | 0.955 | 0.959 | 0.933 |

### 3.6. Adsorption Isotherms

The adsorption capacity of TCS by PLA was about 1.2 times that of PE, and about 3.4–4.0 times as high as the adsorption capacity of PVC and PS, which followed in the order of PLA > PE > PVC > PS. The difference of the TCS adsorption amounts may be attributed to the physio-chemical properties of microplastics. Zuo et al. [10] found that the adsorption rates of poly (butyleneadipate-co-terephthalate) were higher than PE and PS for phenanthrene. Based on the glass transition temperatures, three biodegradable microplastics can be divided into rubbery microplastics (PE) and glassy microplastics (PVC and PS). The former has large amounts of unoccupied pores between molecules, while the latter have a tight structure and congested internal free volume [28,29]. Thus, the rubbery structure of microplastics leads to a higher adsorption capacity of PE than PS and PVC. Wang et al. [25] explored the adsorption of FOSA on non-biodegradable PE, PVC and PS, which supporting that PE exhibits larger sorption capacity than PS and PVC. The stronger loading of TCS on biodegradable PLA seemed to be related to the oxygen-containing functional groups like –OH and –COOH on the surface of the PLA. The results indicated that the hydrogen bond between chemical groups of PLA and TCS play a crucial role in the potential adsorption mechanism.

To further understand the adsorption behavior and mechanisms, the Freundlich isotherm model, Langmuir isotherm model and linear isotherm model for TCS adsorption on PLA, PE, PVC and PS were fitted and shown in Figure 6. The obtained parameters were given in Table 2. As can be seen from Table 2, the Freundlich model gave the highest $R^2$ values (0.999) for TCS adsorption on PLA, indicating that the heterogeneous chemisorption adsorption might be predominant. The monomolecular adsorption was possibly due to the homogeneous specific adsorption site. The results were consistent with previous reports by Hüffer and Hofmann [29]. The Freundlich model and linear model were fitted well with the PE, PVC and PS adsorption data, which illustrated that the adsorption of TCS by PE, PVC and PS was dominated by the monolayer and multilayer coverage. A similar conclusion was found by Chen et al. and Yu et al. [30,31].

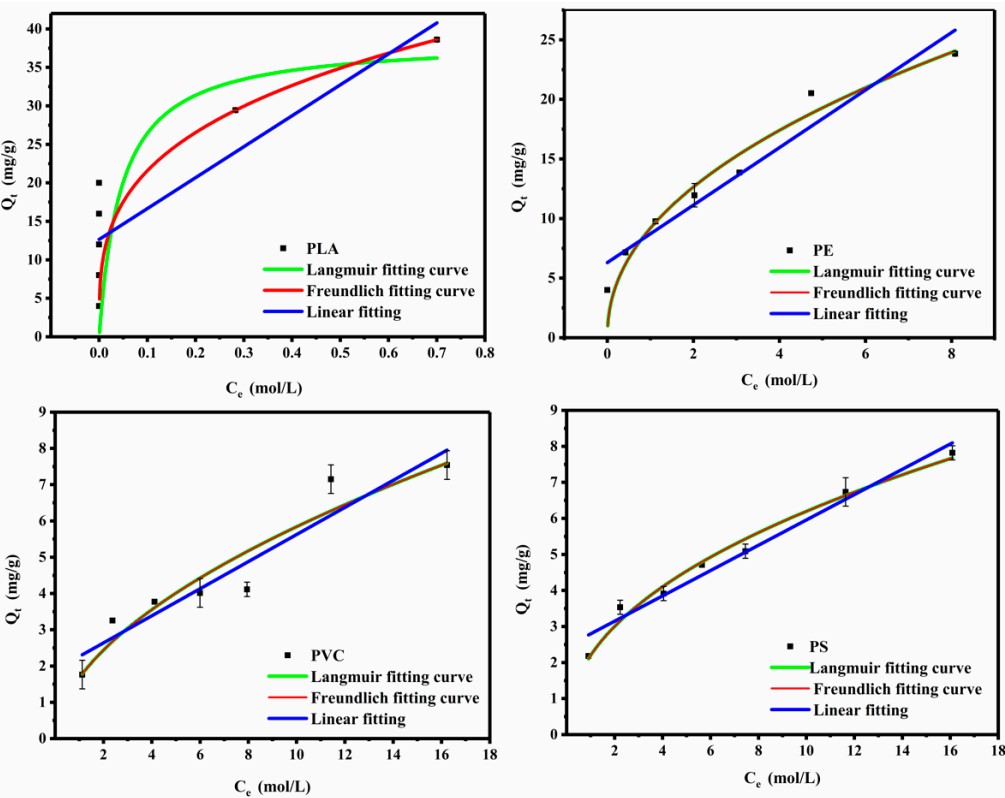

**Figure 6.** The adsorption isotherms of TCS onto PLA, PE, PVC and PS.

**Table 2.** Langmuir, Freundlich and Linear isotherm parameters for TCS adsorption.

| Isotherm Model | Constants | PLA | PE | PVC | PS |
|---|---|---|---|---|---|
| Freundlich model | $K_F$ (mg kg$^{-1}$) (mg L$^{-1}$)$^{-n}$ | 42.9 | 9.19 | 1.68 | 2.20 |
| | $n$ | 0.300 | 0.460 | 0.542 | 0.450 |
| | $R^2$ | 0.999 | 0.967 | 0.909 | 0.984 |
| Langmuir model | $K_L$ (L mg$^{-1}$) | 21.7 | 0.354 | 0.113 | 0.196 |
| | $Q_m$ (mg g$^{-1}$) | 38.6 | 31.3 | 11. 4 | 9.64 |
| | $R^2$ | 0.528 | 0.767 | 0.709 | 0.784 |
| Linear model | $K_d$ (L mg$^{-1}$) | 40.2 | 2.41 | 0.373 | 0.351 |
| | $R^2$ | 0.780 | 0.940 | 0.900 | 0.971 |

## 4. Conclusions

A series of studies, including influence factors (e.g., particle sizes, dosage, pH and salinity), adsorption kinetics and isotherms were performed to evaluate the vector for TCS by biodegradable PLA compared with that of non-biodegradable PE, PVC and PS. The effect of the size of biodegradable PLA and non- biodegradable PE, PVC and PS on adsorption capacity of TCS was not obvious. The microplastics dosage had a positive effect on the removal efficiency of TCS. The solution pH of 4, 7 and 10 had a certain impact on the adsorption of TCS on microplastics forced by electrostatic interaction. The adsorption capacity of TCS by non-biodegradable PE, PVC and PS decreased as NaCl concentrations increased. However, the adsorption variation for PLA at a different salinity was not noticeable. In this study, the results demonstrated that biodegradable PLA has a high adsorption capacity for TCS. The hydrogen bond formed by the oxygen-containing functional groups on PLA and TCS was predominate in adsorption. The present work reveals that biodegradable PLA can act as an important vector for TCS and cause potential environmental risks. Further studies are needed to identify the mechanism of the TCS adherence to PLA and to understand the desorption risks by adsorbed TCS on PLA.

**Supplementary Materials:** The following supporting information can be downloaded at: https://www.mdpi.com/article/10.3390/su141912872/s1, Figure S1: The particle sizes distribution of PLA, PE, PVC and PS; Figure S2: Species distribution of TCS as a function of pH values; Table S1: The molecular structure of PLA, PE, PVC and PS; The measure method of point of zero charge (PZC).

**Author Contributions:** Conceptualization, Z.L.; methodology, Z.L.; validation, Z.L.; formal analysis, Z.L.; investigation, Z.L.; resources, L.X.; data curation, Z.L.; writing—original draft preparation, Z.L.; writing—review and editing, Z.L. and L.X.; visualization, Z.L.; supervision, L.X. All authors have read and agreed to the published version of the manuscript.

**Funding:** This research received no external funding.

**Institutional Review Board Statement:** Not applicable.

**Informed Consent Statement:** Not applicable.

**Data Availability Statement:** Not applicable.

**Conflicts of Interest:** The authors declare that they have no conflict of interest.

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
