# Peer review of "Can Polylactic Acid (PLA) Act as an Important Vector for Triclosan?"

_sustainability, doi:10.3390/su141912872_

Round 1

Reviewer 1 Report

The manuscript describes the adsorption of triclosan to a biodegradable microplastic and three non-biodegradable microplastics. Overall, I didn’t find the manuscript suitably novel for publication. There are now numerous studies showing the same or very similar results.

Include numerical data in abstract

There are several sentences that are poorly written or don’t make sense e.g. “plastic products has been deeply…” The entire manuscript needs proofread

L100 and L112 – what was the wavelength. You seem to contradict yourself

L141 – please check the units of Kf

L146-155 – please explain the results fully in the text

Figure 2 – what do the error bars show? You only did the experiments in duplicate

8 figures and 3 tables is too much and needs reduced

Author Response

Please see the attachment of "Respose to reviewer 1 comments"

Reviewer 2 Report

Title:  Can Polylactic acid (PLA) act as an important vector for Triclosan?

The manuscript demonstrated that biodegradable microplastics (PLA) could bring potential risks to the environment.

The topic is of high interest.  However, it could be published after consideration of the following points.

1) In the Introduction, the evolution and use of PLA in different (in table form) could be interesting for improving the interest of academics and companies. Examples are:

https://doi.org/10.1016/j.addr.2016.04.003

https://doi.org/10.1002/ghg.1853

https://doi.org/10.1016/j.addr.2016.06.014

2) The conclusions are too short. It is better to add the section on future trends. 

Author Response

Please see the attachment of "Respose to reviewer 2 comments"

Reviewer 3 Report

1) Check the representation of TCS in Line 16
2) Line 72: Statement is contradicting with novelty of work. What is the purpose of including biodegradable microplastic in this work?
3) How solid to solvent ratio was fixed at 15 mg/30 mL?
4) Why experiments were not performed in triplicate?
5) Why TCS concentrations were determined by UV-Vis Spec and HPLC?
6) Why there is no mention about extraction phase in HPLC protocol?
7) What type of analysis was performed for kinetics and isotherm, linear or non-linear?
8) Y-axis of Figure 1 needs elaboration. The readers may confuse % with % removal.
9) Why dosage 2.5 mg/L is not included in Figure 3?
10) Why only Qt is represented in Figure 2 and % removal in Figure 3? Why both were not represented in all parameteric effects?
11) How PSO model was concluded from kinetic studies? Both show R2 value > 0.9.
12) Comncluding remark is not satisfactory.
13) Table numbering should be different for main document and supplemengtary material. Table 1 is missing in main document.
14) Figures were not cited in main document. Figure S1 is not available in supplementary material.
15) Equations were not cited in main document.
16) Video S1 is not available in supplementary material.
17) References after 2020 may be added.

Author Response

Please see the attachment of "Respose to reviewer 3 comments"

Reviewer 4 Report

This work focuses on investigation the Polylactic acid (PLA) ability to act as vector for Triclosan. The influence factors of particle size, dosage, pH and salinity affecting the adsorption capacity were discussed in detail. This study contributes to a better understanding of the adsorption kinetics and isotherm of biodegradable microplastics for Triclosan. The manuscript is well prepared and technically sound, and these findings are important and meaningful. 

 Special comments

1. Page 2 Line 50, "Plastics" should be changed to “Microplastics”.
2. Page 2 Line 51, Change “Biodegradable plastics” to “Biodegradable microplastics”.

3. Page 2 Line 56, this sentence should be reorganized.
3. Page 3 Line 100, "at 282nm by UV-visible spectrophotometer" is ambiguous expression.
4. Page 4 Line 146, “3.1 …Particle”, The letter P should be lowercase.

4. Page 4 Line 152-153, authors should organize the sentence again.
5. Page 5, Line 168-169, The sentence should be checked again.
6. In my opinion, the measure method of point of zero charge (PZC) should be provided in Supplementary materials.
7. Page 6, The title of the vertical axis of Figure 5 needs to be verified.
8. Page 10, Line 275, "…behaviors are… " should be changed to “…behavior is…”.
9. The reference list should be checked over to meet the publication requirements.

10. Please add some important references.

Author Response

Please see the attachment of "Respose to reviewer 4 comments"

Round 2

Reviewer 1 Report

Unfortunately the author failed to comment on, or address the more fundamental points I made in my previous review:

Overall, I didn’t find the manuscript suitably novel for publication. There are now numerous studies showing the same or very similar results.

This needs addressed before being considered for publication.

Author Response

Thanks for this helpful suggestion, and we appreciate the reviewer's point of view is unique. We agree that there are some published papers about microplastics adsorption pollutants, as we have mentioned in the introduction part: “However, the knowledge of adsorption affinity of PLA is limited.” “The high accumulation of environmental pollutants on non-biodegradable microplastics discussed in previous studies [15-16].” We took efforts to collect literature by using several sources of database including Web of Science, Elsevier and Springer. Keywords used in searches of different sections include: “triclosan”, “polylactic acid (PLA)”, “vectors”, “carriers”, and so on. After removal of duplicates and being filtered by title and abstract, two studies that cover adsorption triclosan by non-biodegradable and biodegradable microplastics remain (Webb et al., 2020; Shi et al., 2022). However, the study by webb et al. mainly focused on the adsorption of triclosan by non-biodegradable PE (Webb et al., 2020). The influence of particle size and dosage were not investigated in PLA adsorption TCS by Shi et al. (Shi et al., 2022), and the interaction of biodegradable PLA with TCS remains unclear. It is of paramount significance to conduct a series of studies evaluating the adsorption vector for TCS by PLA. The objectives of this study are: (1) to analyse and compare the PLA ability to act as vector for TCS; (2) to demonstrate the influence factors of particle size, dosage, pH and salinity affecting the adsorption capacity; (3) to investigate adsorption kinetics and isotherm revealing the adsorption behavior.

We revised and emphasized the research significance and innovation points in the paper.

The newly added literatures were listed in the reference as follows:

Webb, S.; Gaw, S.; Marsden, I.D.; McRae, N.K. Biomarker responses in New Zealand green-lipped mussels perna canaliculus exposed to microplastics and triclosan. Ecotoxicol. Environ. Saf. 2020, 201, 110871.

Shi, K.; Zhang, H.; Xu, H.M.; Liu, Z.; Kan, G.F.; Yu, K.; Jiang, J. Adsorption behaviors of triclosan by non-biodegradable and biodegradable microplastics: Kinetics and mechanism. Sci. Total Environ. 2022, 842, 156832.

Reviewer 3 Report

Authors answered queries of reviewer. Hence, I recommend the manuscript for publication in its present form.

Author Response

Point 1: Authors answered queries of reviewer. Hence, I recommend the manuscript for publication in its present form.

Response 1: Thank you for the positive comments on our manuscript.

Round 3

Reviewer 1 Report

none